# Exploratory Circular RNA Profiling in Adrenocortical Tumors

**DOI:** 10.3390/cancers14174313

**Published:** 2022-09-02

**Authors:** Péter István Turai, Gábor Nyirő, Katalin Borka, Tamás Micsik, István Likó, Attila Patócs, Peter Igaz

**Affiliations:** 1Department of Endocrinology, ENS@T Research Center of Excellence, Faculty of Medicine, Semmelweis University, H-1083 Budapest, Hungary; 2Department of Internal Medicine and Oncology, Faculty of Medicine, Semmelweis University, H-1083 Budapest, Hungary; 3MTA-SE Molecular Medicine Research Group, Eötvös Loránd Research Network, H-1083 Budapest, Hungary; 4Department of Laboratory Medicine, Faculty of Medicine, Semmelweis University, H-1089 Budapest, Hungary; 5Department of Pathology, Forensic and Insurance Medicine, Semmelweis University, H-1091 Budapest, Hungary; 6Department of Pathology and Experimental Cancer Research, Semmelweis University, H-1085 Budapest, Hungary; 7MTA-SE Hereditary Tumors Research Group, Eötvös Lóránd Research Network, H-1089 Budapest, Hungary; 8Department of Molecular Genetics, National Institute of Oncology, H-1122 Budapest, Hungary

**Keywords:** adrenocortical carcinoma, adrenocortical adenoma, adrenal tissue, circular RNA, biomarker, expression, microRNA, RT-qPCR, next-generation sequencing

## Abstract

**Simple Summary:**

The histological differential diagnosis of adrenocortical adenoma and carcinoma is difficult and requires great expertise. Measures taken towards the distinction of adrenal tumors are of paramount importance. The non-coding circular RNAs (circRNAs) were shown to be expressed in a tissue and tumor specific manner. CircRNAs are investigated as a useful adjunct to the differential diagnosis of benign and malignant tumors of several organs, but they have not been investigated in adrenocortical tumors yet. Here, we have performed circRNA profiling in adrenocortical tumors by next-generation sequencing to detect already known and de novo circRNAs. Out of the five most differentially expressed circRNAs, *circPHC3* could be confirmed by TaqMan RT-qPCR to be overexpressed in carcinoma and adenoma vs. healthy tissues in an independent validation cohort.

**Abstract:**

Differentiation of adrenocortical adenoma (ACA) and carcinoma (ACC) is often challenging even in the histological analysis. Circular RNAs (circRNAs) belonging to the group of non-coding RNAs have been implicated as relevant factors in tumorigenesis. Our aim was to explore circRNA expression profiles in adrenocortical tumors by next-generation sequencing followed by RT-qPCR validation. Archived FFPE (formalin-fixed, paraffin embedded) including 8 ACC, 8 ACA and 8 normal adrenal cortices (NAC) were used in the discovery cohort. For de novo and known circRNA expression profiling, a next-generation sequencing platform was used. CIRI2, CircExplorer2, AutoCirc bioinformatics tools were used for the discovery of circRNAs. The top five most differentially circRNAs were measured by RT-qPCR in an independent validation cohort (10 ACC, 8 ACA, 8 NAC). In silico predicted, interacting microRNAs potentially sponged by differentially expressed circRNAs were studied by individual RT-qPCR assays. We focused on overexpressed circRNAs here. Significantly differentially expressed circRNAs have been revealed between the cohorts by NGS. Only *circPHC3* could be confirmed to be significantly overexpressed in ACC, ACA vs. NAC samples by RT-qPCR. We could not observe microRNA expression changes fully corresponding to our sponging hypothesis. To the best of our knowledge, our study is the first to investigate circRNAs in adrenocortical tumors. Further studies are warranted to explore their biological and diagnostic relevance.

## 1. Introduction

According to estimations of high-resolution abdominal imaging studies, adrenal tumors can be revealed in every 25th person during their lifetime [1]. These are mostly benign (49–69%), but rarely (incidence: 0.7–2/million/year) adrenocortical carcinoma (ACC) occurs with a rather dismal prognosis (5 year survival less than 30% in advanced stages) [2,3,4]. The adrenolytic mitotane is the only available adrenal tumor specific treatment in ACC [5,6]. While the prevalence of ACC among adrenal incidentalomas is low, around 13% of ACCs are diagnosed as incidentalomas [7]. The preoperative, but even the histological differential diagnosis of adrenocortical adenoma (ACA) and carcinoma is challenging [8].

Several biomarkers are being investigated to help the differential diagnosis of ACA and ACC both as tissue and circulating markers [9,10]. Non-coding RNA molecules (ncRNA) have been found to be promising in several studies including microRNAs and long non-coding RNA molecules [11,12]. The group of ncRNA molecules include several members that can be categorized as constantly expressed housekeeping RNAs, e.g., ribosomal RNA (rRNA) or transfer RNA (tRNA), and as regulatory RNAs, such as microRNAs (miRNA, miR, 19–25 nucleotides), long non-coding RNA (lncRNA, more than 200 nucleotides) and circRNAs [13].

Initially, circular RNAs (circRNA; a covalently closed RNA molecule) discovered more than 30 years ago in mammalian cells, were not associated with substantive biological functions, and were considered to be defects of the normal splicing mechanism, so-called trans-splicing resulting in “scrambled exons” [14].

Recent discoveries have revealed that human genes are capable of expressing thousands of circular RNAs, and indeed 5.8–23% of the end products of expression of the active genes are circRNA [15]. Despite the observation that many of them contain exons, proteins are rarely formed from them [16]. circRNAs control gene expression without altering the very sequence of DNA mainly on a post-transcriptional level achieved by epigenetic modulation [17]. CircRNAs are formed from pre-mRNA (not yet fully mature form containing introns) during a process called “backsplicing”, whenever a covalent bond is formed between the 3′ and the 5′ ends [18]. Thus, it forms a so-called back splice junction (BSJ) where donor sequence of an upstream exon covalently links to the acceptor sequence of a downstream exon. Identification of BSJs is crucial when validating circRNAs with so-called divergent qPCR primers [19]. Divergent primers designed to span the circRNA backsplice junction sequence and oriented to amplify away from each other in a genomic context can specifically amplify the circRNAs without amplifying the counterpart linear RNA [19]. The degree of expression of circRNAs is different from the expression of their linear counterpart RNA. The explanation lies in the different backsplicing frequency primarily determined by the intron’s structure (repetitive or reverse complementary sequences), which renders the expression of circRNAs independent of their carrier gene’s expression [18]. Regarding their biological function, circRNAs are implicated in the regulation of neural function, innate immunity, cell proliferation as well as in the pathogenesis of various solid (colorectal cancer, hepatocellular cancer, lung cancer, glioma, osteosarcoma, etc.) and hematological tumors [20]. Differential expression of circRNA have been described in benign and malignant tumors of various organs, suggesting their utility as markers of malignancy [21,22,23,24], but to the best of our knowledge, no circRNA profiling has been performed in adrenocortical tumors, yet.

The molecular activities of circRNA include (a): the processing of circRNAs affects splicing of their linear mRNA counterparts; (b): circRNAs can regulate transcription of their parental genes; (c): circRNAs can regulate splicing of their linear cognates; (d): circRNAs can act as miRNA sponges; (e): circRNAs can act through associated proteins; (f): circRNAs can be translated, (g): circRNAs containing IRES (internal ribosome entry) are resources for derivation of pseudogenes, which are DNA sequences that resemble a gene but have been mutated into an inactive for form, not resulting in functional proteins [16]. Unlike linear mRNAs, circRNAs lack poly(A)-tail and free end, thus rendering them resistant to RNase digestion, especially to RNase R exonuclease, which can be exploited in circRNA studies [25,26,27].

Here, we report the first circRNA profiling in adrenocortical tumors using NGS.

## 2. Materials and Methods

### 2.1. Tissue Collection and Ethics Approval

A total of 18 ACC, 16 ACA and 16 normal adrenal cortex (NAC) formalin-fixed paraffin-embedded (FFPE) samples were included in the study. NAC samples were obtained from patients undergoing total nephrectomy for kidney tumors. Samples were histologically confirmed by two different adrenal expert pathologists and only definite parts of the blocks were dissected for the study. The Weiss-score formed the basis for the diagnosis of adrenocortical malignancy but clinical presentation and hormonal activity were also considered during diagnosis [3]. The discovery cohort was comprised of 8 ACA, 8 ACC, 8 NAC and the independent validation cohort contained another 10 ACC and 8 ACA 8 NAC FFPE samples. We used the same validation cohort for the miRNA expression analysis. The basic characteristics of the patients is provided in Table 1.

All experiments were performed in accordance with pertinent guidelines and regulations, and informed consent was also obtained from the patients involved. The study was approved by the Ethical Committee of the Hungarian Health Council (56720-2/2018/EKU).

### 2.2. Sample Processing and RNA Isolation

RecoverAll Total Nucleic Acid Isolation Kit for FFPE (Catalog Number: AM1975, Thermo Fisher Scientific, Waltham, MA, USA) was used for total RNA isolation. In the validation cohort, as a spike-in control for isolation efficiency 1 μL of 0.002 fmol/µL syn-cel-miR-39-3p was used according to the manufacturer’s protocol for miRCURY LNA RNA Spike-in Kit (Qiagen GmbH, Hilden, Germany, Catalog Number: 339390) and was added before the nucleic acid isolation step. Total nucleic acid quantity was measured by NanoDrop 2000 Spectrophotometer (Thermo Fisher Scientific, Waltham, MA, USA) after isolation and for RNA only Qubit 4 Fluorometer with Qubit™ hsRNA Assay Kit (Thermo Fisher Scientific, Waltham, MA, USA) was used. Total RNA was stored at −80 °C until further processing. Assessment of RNA quality was carried out by Agilent 2100 Bioanalyzer (Agilent Technologies, Santa Clara, CA, USA) with RNA 6000 Nano and Pico kits before next-generation sequencing step.

### 2.3. RNase R Treatment, Polyadenylation, and Poly(A)+ RNA Depletion (RPAD)

After the isolation step, the total RNA sample contained various RNA molecules, e.g., rRNA, mRNA, snRNA, miRNA, and circRNA. Selective degradation of linear RNAs was implemented by 1 μL RNase R (20 U/μL) (Lucigen, Epicentre) in a 2 μg RNA containing 20 μL mix with 2 μL 10X RNase R reaction buffer (provided with RNase R), 1 μL RiboLock (Thermo Fisher Scientific, Waltham, MA, USA) followed by 30 min incubation at 37 °C.

After digestion, purification was applied according to the miRNeasy Kit (QIAGEN) protocol, and the purified solution was eluted in 40 μL of nuclease-free water.

Beyond circRNAs, RNase R does not digest, e.g., double-stranded or highly structured RNAs. We used Poly(A) tailing Kit (Thermo Fisher Scientific, Waltham, MA, USA) to deplete any non-circRNAs for maximal purification. A 40-μL polyadenylation reaction mix was prepared according to the protocol with 20 μL of RNase R-treated RNA, 4 μL of 10X poly(A) polymerase buffer, 1 μL of E-PAP poly(A) polymerase (provided with Poly(A) tailing Kit), 4 μL of 10 mM ATP solution (provided with Poly(A) tailing Kit), 4 μL of 25 mM MnCl2 (provided in the Poly(A) tailing Kit) and 1 μL of RiboLock RNase inhibitor (Thermo Fisher Scientific, Waltham, MA, USA). Subsequent incubation lasted 30 min at 37 °C.

During the polyadenylation process, we dissolved 10 μL of well-suspended oligo-dT magnetic beads (provided with Poly(A)Purist™ MAG Kit) with 100 μL of 1X binding buffer (provided with Poly(A)Purist™ MAG Kit) and incubated on the magnetic stand for 1 min. Supernatant was discarded and the step was repeated two additional times to fully activate the beads. We dissolved the active beads in 40 μL of 2X binding buffer (provided with Poly(A)Purist™ MAG Kit) and added to the 40-μL polyadenylation reaction. Next, the polyadenylation reaction was incubated with the magnetic beads for 5 min at 75 °C, and that was followed by incubation for 20 min at 25 °C to allow the oligo-dT beads to bind the poly(A)-tailed RNAs. Then, the reaction was put on a magnetic stand for 1 min to collect and proceed with the supernatant. Again, purification was applied according to the miRNeasy Kit (QIAGEN) protocol, and the purified solution was eluted in 40 μL of nuclease-free water.

### 2.4. Next-Generation Sequencing (NGS)

The cDNA library was made from purified circRNA solution by using NEBNext Ultra II RNA Library Prep Kit for Illumina E7770 (New England Biolabs, Ipswich, MA, USA) according to the instructions of the manufacturer. The prepared library was quantified by NebNext Library Quantification Kit for Illumina. Fragmentation sizing was carried out by High Sensitivity dsDNA chip on an Agilent 2100 Bioanalyzer (Agilent Technologies, Santa Clara, CA, USA). Next-generation sequencing was performed by MiSeq Reagent Kit v3 600 on Illumina MiSeq instrument (Illumina, San Diego, CA, USA).

### 2.5. Bioinformatics Analysis of circRNA

RNA-Seq Quality control and alignment: After adapter and quality trimming (Q > 30), the reads were mapped to the reference genome hg38 using the Burrow-Wheeler Aligner (BWA-MEM) [28]. For visualization of the intersect values of different tools and biological groups, the VennDiagramm R package was used [29].

CircRNA identification: Three different circRNA detection tools were tested, AutoCirc [30], CIRI2 [31] and CircExplorer2 [32]. Autocirc uses unmapped read information from the mapping files, uses bowtie2 aligner to map 20bp anchors from both ends of the read to detect back-spliced junction (BSJ) with unambiguous GT-AG breakpoint detection, and identifies BSJs that meet the annotated exon boundaries [30]. CIRI2 uses Maximum-likelihood estimation based algorithm to detect BSJs from the mapping files with multiple seed matching [31]. CircExplorer2 has two main pipelines: the annotation pipeline for detecting circRNAs with fusion junction annotation of known gene boundaries, and the characterization pipeline for de novo circRNA assembly and characterization of alternative backsplicing and alternative splicing [32]. Based on performance (number of detected circRNA, number of known circRNAs, realignment of fusion reads) circExplorer2 results were chosen for downstream analysis. Annotation, and the full sequences of circRNA, had been obtained by circRNAprofiler [33]. With the circRNAprofiler function getBackSplicedJunctions(), a count matrix was generated from circExplorer2 data with the raw count value for all circRNA identified in each sample.

CircRNA expression: Differentially expressed circRNAs were identified using the limma-trend algorithm [34] with TMM normalisation method, and a prior count of 1. P-value adjustment was calculated with Benjamini–Hochberg FDR method. For heatmap visualization, the z-score values were calculated for expression values (to log2-counts-per-million, logCPM).

### 2.6. CircRNA Validation by RT-qPCR

The RT-qPCR validation of NGS results was performed on a separate validation set of samples. First, selective degradation of linear RNAs was implemented by adding 1 μL RNase R (20 U/μL) (Lucigen, Epicentre) in a 2 μg RNA containing 20 μL mix with 2 μL 10X RNase R reaction buffer (provided with RNase R), 1 μL RiboLock (Thermo Fisher Scientific, Waltham, MA, USA) followed by 30 min incubation at 37 °C.

After RNase R reaction, purification was applied according to the miRNeasy Kit (QIAGEN) protocol, and the purified solution was eluted in 40 μL of nuclease-free water.

Then, remaining RNA was reverse-transcribed using the Maxima H Minus First Strand cDNA Synthesis Kit (Catalog Number: K1652 Thermo Fisher Scientific, Waltham, MA, USA) according to the manufacturer’s protocol with random (Roche, Basel, CH, Switzerland).

TaqMan miRNA divergent primer assays (Catalog Number: 230744765) were used for the RT-qPCR validation of the top five differentially expressed circRNAs (*PHC3*:-:chr3: 170145422:170149244; *FCGBP*:-:chr19: 39877662:39893527; *TIMMDC1*:+:chr3: 119503531:119504021; *KDM4C*:+:chr9: 6880011:6893232; *MAN1A2*:+:chr1: 117402185:117420649). For divergent circRNA primer and probe sequences, see Appendix A.

As external spike-in control synthetic cel-miR-39 (ID: 000200) was used. Quantitative real-time PCR was performed with the TaqMan Fast Advanced Master Mix (Catalog Number: 4444557; Thermo Fisher Scientific) on a Quantstudio 7 Flex Real-Time PCR System (Thermo Fisher Scientific, Waltham, MA, USA) according to the manufacturer’s protocol for TaqMan assays. Samples were always run triplicated. Negative control reactions contained no cDNA templates. Positive control reactions contained human genomic DNA and 100× diluted cDNA of the NGS library. For data evaluation, the dCt method (delta Ct value equals target RNA’s Ct minus external control RNA’s Ct) was used by Microsoft Excel (Microsoft, Redmond, WA, USA). Statistical analyses (ANOVA (analysis of variance) and Tukey post-hoc tests) were performed using GraphPad Prism 7.

### 2.7. In Silico Prediction of Potential circRNA-miRNA Interactions

Using clipSearch [35], we searched for miRNA targets for PHC3 circRNA. The mature_high_conf_v22_1.fa miRNA sequences were used as targets. The results were obtained as a text file. To further evaluate and filter the results, we extracted the target list table with the related parameters.

Three potential miRNAs have been selected that were already described as relevant in adrenocortical tumors based on the literature data. *Hsa-let-7c* as well as *hsa-miR-214* and *hsa-miR-195* were shown to be down-regulated in ACCs compared with ACAs [36,37,38,39,40,41,42].

### 2.8. miRNA Analysis by RT-qPCR

We used the same validation cohort for the miRNA expression analysis.

A 2-step RT-qPCR method was used. Each sample was processed separately for all miRNA targets. A total of 10 nanograms of isolated total RNA was used in individual RT reactions.

First, TaqMan miRNA Reverse Transcription Kit (Catalog Number: 4366596, Thermo Fisher Scientific, Waltham, MA, USA) and individual TaqMan MiRNA Assay primer mixes (Catalog Number: 4427975, Thermo Fisher Scientific, Waltham, MA, USA) were used for reverse transcription of total RNA. The expression of *hsa-miR-195* (ID: 000494), *hsa-let-7c-3p* (ID: 002479) and *hsa-miR-214* (ID: 002306) were measured, and as an internal control *RNU48* (ID: 001006) along with spike-in control *cel-miR-39* (ID: 000200) as an external control were used.

For quantification, TaqMan Fast Advanced Master Mix (Catalog Number: 4444963, Thermo Fisher Scientific, Waltham, MA, USA) was used on a QuantStudio 7 Flex Real-Time PCR System (Thermo Fisher Scientific, Waltham, MA, USA) according to the manufacturer’s protocol. Negative control reactions contained no cDNA templates, and all samples were measured in triplicate. A total of 0.67 µL of undiluted cDNA was used as template. For data evaluation, we used the ΔΔCt method [43]. Statistical analyses were performed using GraphPad Prism 7.

## 3. Results

### 3.1. Identification of Differentially Expressed circRNAs

NGS was used to identify differentially expressed circRNAs in the discovery cohort comprised of 8 ACC, 8 ACA and 8 NAC samples. A total of 6532 already known and de novo circRNAs were found by the three detection tools used, and out of them 445 circRNAs were found consistently with all three tools (Figure 1). A total of 3912 circRNAs were already known according to CircAtlas [44]. The list of all circRNAs found is shown in Appendix A.

Figure 2 presents a heatmap showing the variation of circRNA expression levels between ACC vs. ACA samples. Polyhomeotic-like protein 3 (*PHC3*) circRNA was identified with Log2 Fold Changes of greater than 1.0. The top 10 differentially expressed circRNAs between ACC vs ACA are provided in Appendix A.

### 3.2. Validation of Differentially Expressed circRNAs

Considering that the expression level of down-regulated circRNAs was very low and hard to detect, we only focused on the up-regulated circRNAs. Based on the heatmap, we first selected the five top up-regulated circRNAs showing the highest expression differences. For heatmap visualization, the z-score values were calculated for expression values (log-count-per-million, logCPM). (Appendix A). To verify the circRNA-seq results of these five circRNAs, RT-qPCR was performed with divergent primers spanning backsplice junctions on an independent validation cohort of 10 ACC, 8 ACA and 8 NAC FFPE samples. As shown in Figure 3, *circPHC3* (CircAtlas ID: *hsa-PHC3_0006;* CircPedia ID: *HSA_CIRCpedia_44334*) was significantly up-regulated in ACC and ACA patients compared to NAC patients, which were consistent with our previous sequencing results.

### 3.3. In Silico Prediction of Potential circRNA-microRNA Interactions (Sponging)

The results of the in silico prediction of potential microRNAs sponged by *circPHC3* using clipSearch are presented in Appendix A.

### 3.4. Putative miRNA Interaction with circPHC3

In silico predicted microRNA sponging of differentially expressed circRNA was studied by individual RT-qPCR assays. Three potential miRNAs have been selected. *Hsa-let-7c-3p* and *hsa-miR-214* were shown to be significantly down-regulated in ACC vs NAC samples (Figure 4A,B). *Hsa-miR-195* was significantly down-regulated in ACC vs ACA samples (Figure 4C).

## 4. Discussion

The histological differential diagnosis of adrenocortical tumors is difficult and requires great expertise. Novel biomarkers to differentiate between the two entities are intensively sought for.

Circular RNAs are even more stable than microRNAs due to their circular structure and resistance to RNase, but they are less abundant. Several studies have reported the differential expression of circRNA in various tumors and also suggested their use as molecular markers, e.g., in prostate cancer, lung cancer, and breast cancer [21,22,23,24,45,46,47,48].

Circular RNAs have not yet been investigated as markers of adrenocortical malignancy and, to the best of our knowledge, we have performed large scale, NGS-based circular RNA profiling in adrenocortical tumors for the first time. We have found only a single in vitro study on circular RNA in ACC to date examining a single circRNA [49].

Despite having found a large number of potentially differentially expressed circRNA by NGS and having selected the top five circRNA for validation, we could only validate one circRNA as differentially expressed: *circPHC3* overexpressed in both benign and malignant adrenocortical tumors compared to the normal adrenal cortex. According to our study, it might thus be hypothesized that *circPHC3* could be associated with adrenal tumor formation irrespective of its benign or malignant behavior, and its altered expression could represent a common phenomenon in both benign and malignant adrenocortical tumors [20]. However, based on epidemiological observations, benign-to-malignant progression in the adrenal cortex could be a very rare event, and it is unlikely to account for the vast majority of ACC [50,51,52,53]. Thus, the biological relevance of our observation is unclear.

*CircPHC3* (CircAtlas ID: *hsa-PHC3_0006;* CircPedia ID: *HSA_CIRCpedia_44334*) is transcribed from the *PHC3* locus that encodes Polyhomeotic-like protein 3 and is required to maintain the transcriptionally repressive state of many genes, e.g., Hox genes, throughout development [54]. PHC3 is a member of the human polycomb complex and has been implicated as a tumor suppressor of osteosarcoma [55]. In our study, *circPHC3* can be classified to have oncogenic properties according to the overexpression witnessed in tumorous tissues (both benign and malignant) compared to normal adrenocortical tissues. A possible explanation for the opposite biological functions observed might lie in the tissue and/or tumor specificity of circRNAs analogous to that of microRNAs [56].

CircRNAs act in various manners, and one of their mechanisms of action is related to microRNA sponging of complementary sequences. Based on an in silico prediction, we have selected three miRNAs that could be potentially sponged by *circPHC3*, and due to the overexpression of *circPHC3*, the underexpression of these miRNAs in both benign and malignant tumor tissues relative to the normal adrenal cortex could be expected. *Hsa-let-7c-3p*, a member of the let-7 tumor suppressor miRNA family has already been described to be down-regulated in ACC versus ACA [37]. Moreover, the *let-7* family is identified as a LIN28 regulatory miRNA [57] and LIN28 protein expression was associated with ACC recurrence [58]. *Hsa-miR-214* was previously shown to be down-regulated in ACC vs. adenoma tissues [36]. In HeLa cells, inhibition of *hsa-miR-214* resulted in decreased apoptosis [59]. Both *hsa-let-7c-3p* and *hsa-miR-214* showed lower expression in ACC vs. NAC that could correspond to the sponging hypothesis, but despite some tendency of underexpression in ACA (Figure 4A,B), no significant difference could be found between ACA and NAC. *Hsa-miR-195* was significantly down-regulated in ACC samples compared to ACA samples, corresponding to its tumor suppressor role described in previous studies [37,38,39,40,41,42]. It can thus be concluded that our miRNA expression results are not fully compatible with the in silico predicted potential sponging of these miRNA by *circPHC3*.

Our study has limitations, including the small sample size and the use of FFPE samples. Fresh frozen samples appear to be more suitable for circRNA analysis due to their higher yields [60], and further circRNA profiling studies on adrenocortical tumors could be proposed by using fresh frozen samples. As there is no established set of housekeeping circRNAs in circRNA gene expression studies, we used only external *cel-miR-39* control for validation of *circ-PHC3*. However, in gene expression analysis, the use of external controls is generally supported much better than not using any at all [61]. It must be noted that there are non-laboratory-based publications indeed, where statistical estimates are employed to predict mRNA expression levels in tandem with circRNA expression levels by different informatical algorithms [62,63]. It is also possible to measure mRNA levels with and without RPAD treatment solely for methodologic purposes [26,27]. Since the starting material comes from archived FFPE samples, where mRNA levels might be distorted due to degradation or fragmentation and due to the resulting non-linear RNA degradation in consequence of RPAD method, we aimed to acquire laboratory-based data omitting mRNAs and focus solely on circRNAs for future clinical purposes.

## 5. Conclusions

To the best of our knowledge, our study is the first to report high-throughput circRNA sequencing of adrenocortical tumors. The differential expression of *circPHC3* between adrenal tumors (carcinoma and adenoma) versus normal adrenal cortex samples was shown and validated by RT-qPCR. The biological relevance of *circPHC3* warrants investigation in further studies.

## Figures and Tables

**Figure 1 cancers-14-04313-f001:**
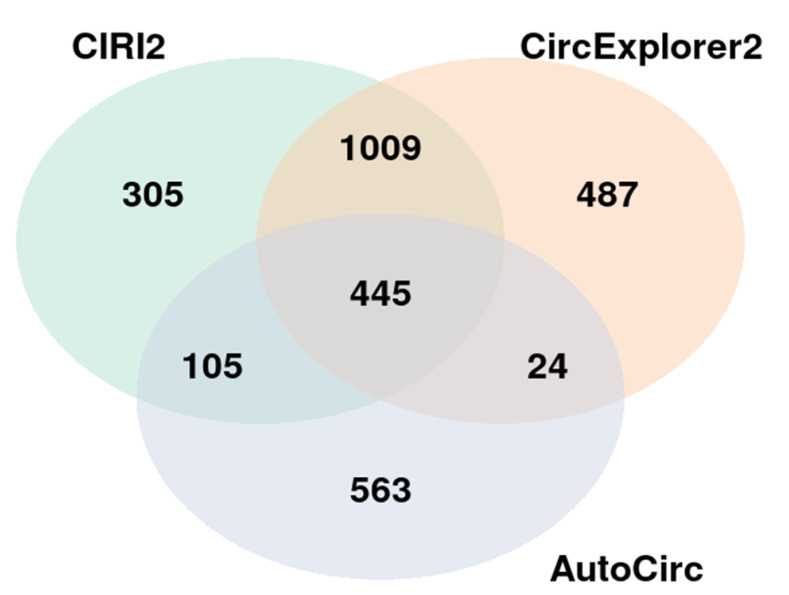
Number of circRNAs found by the three circRNA detecting tools.

**Figure 2 cancers-14-04313-f002:**
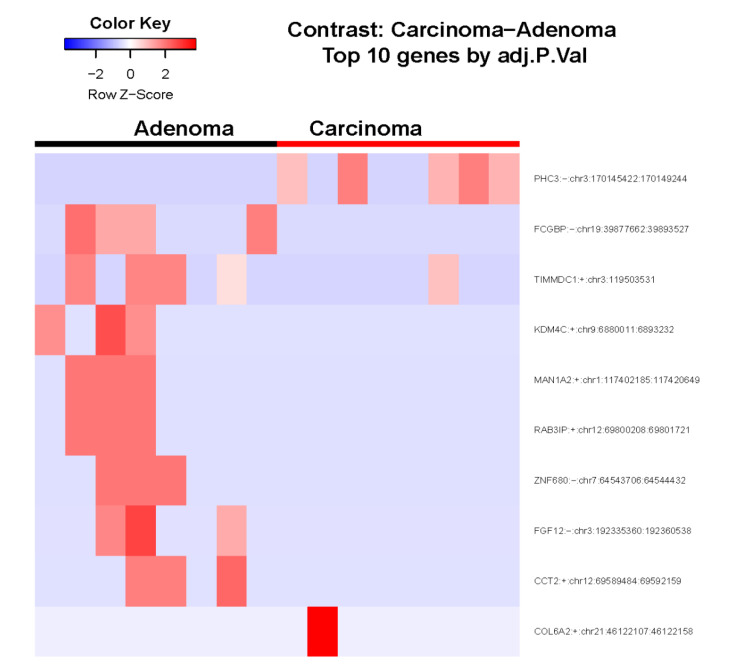
Heatmap showing the variation of circRNA expression levels between ACC vs. ACA samples. Adj.P.Val: adjusted *p*-value.

**Figure 3 cancers-14-04313-f003:**
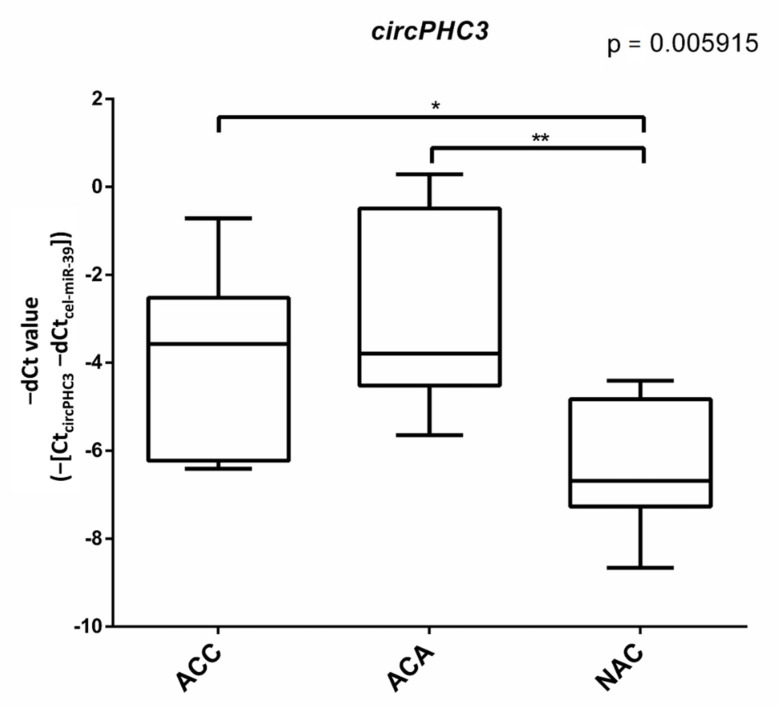
Box plot representing the expression of *circPHC3* in ACA, ACC, and NAC samples. ANOVA (*p* = 0.005915) and Tukey post-hoc tests (*: *p* < 0.05, **: *p* < 0.01) were performed.

**Figure 4 cancers-14-04313-f004:**
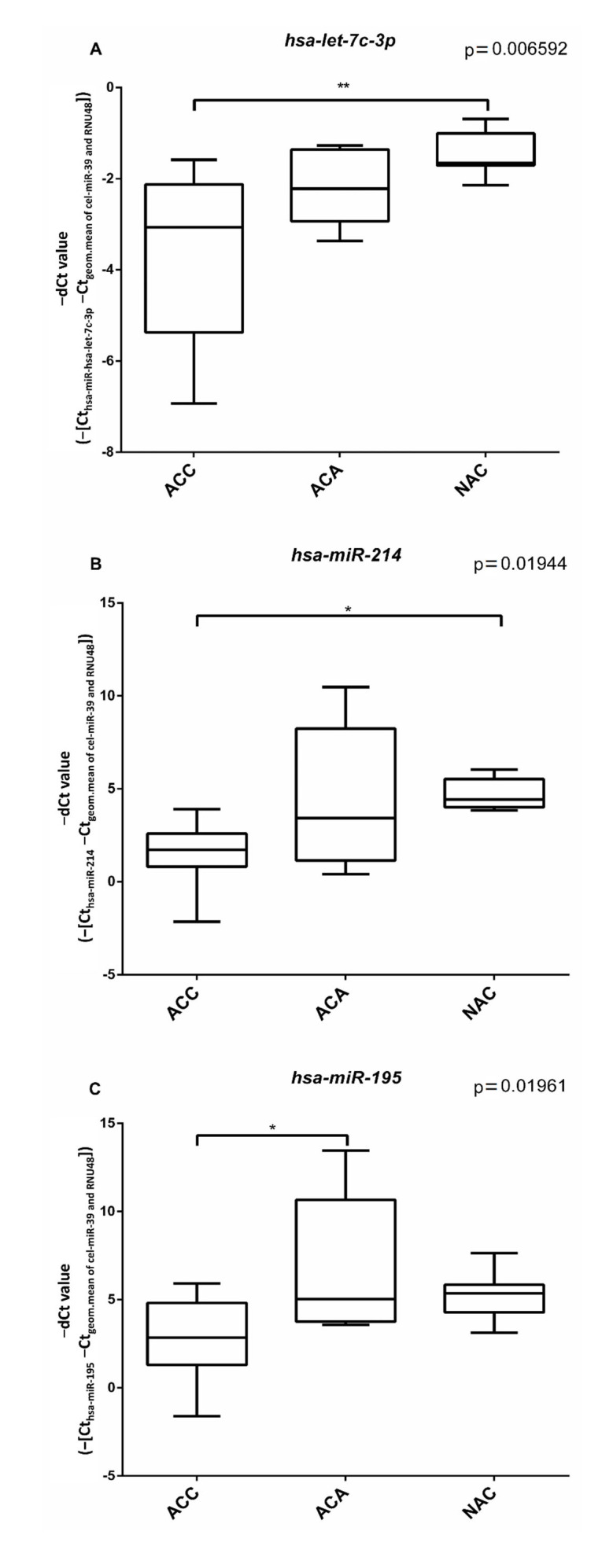
Box plots representing the expression of hsa-let-7c-3p (**A**), hsa-miR-214 (**B**) and hsa-miR-195 (**C**) miRNAs relative to the geometric means of *cel-miR-39* Ct values and *RNU48* Ct values in ACA, ACC and NAC samples. ANOVA and Tukey post-hoc tests (*: *p* < 0.05, **: *p* < 0.01) were performed.

**Table 1 cancers-14-04313-t001:** Basic characteristics of patients. ACA: adrenocortical adenoma; ACC: adrenocortical carcinoma; F: female; M: male; ENS@T: European Network for the Study of Adrenal Tumors; NF: non-functioning; DOC: deoxycorticosterone; DHEAS: dehydroepiandrosterone sulphate.

Cohort/Samples	Mean Age at Sample Taking (Years)	Mean Tumor Size (mm)	Ki-67 (%)	ENSAT Stage	Hormonal Activity
Discovery ACA (7F, 1M)	46	32	-	-	6 cortisol
					2 NF
Discovery ACC (5F, 3M)	39	119	11–18 (2–40)	II:5, III:2; IV:1	1 cortisol 4 NF 1 DOC 1 DOC + cortisol + estradiol 1 cortisol + DHEAS
Validation ACA (7F, 1M)	53	37	-	-	3 NF
					5 cortisol
Validation ACC (6F, 4M)	56	119	18–23 (1–50)	II:1, III:5, IV:4	5 NF
					4 cortisol 1 cortisol + androgen

## Data Availability

The data presented in this study are available in the article and Appendix A.

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
