# Peer review of "Exploratory Circular RNA Profiling in Adrenocortical Tumors"

_cancers, 2022, doi:10.3390/cancers14174313_

Round 1

Reviewer 1 Report

In this paper doctor Turai et al report the results of the evaluation of circRNA profiling in adrenocortical tumors using NGS. The results show that CircRNA are differently expressed and potentially useful to help pathologists differentiate malignant from benign forms. The paper is well written and circRNAs have never been tested in patients with ACC so far. I have no major comments.

MINOR COMMENTS

the authors correctly acknowledge that the low number of patients is the main limitation of this paper however despite this I'm wondering to know if cirRNA are otherwise espessed in ACC patients according to the presence or absence of hormone hypersecretion, if they correlate with the expression of ki67 and disease stage and if they have a potential prognostic value.

Author Response

Please see the attachment for the answer. We thank Reviewer 1 for these positive comments.

Reviewer 2 Report

General comment:

The authors explored ACC-specific circRNAs, which can be used as potential markers distinguishing ACCs from ACAs. They employed the discovery and the validation cohorts including ~10 ACCs samples each. They focused on over-expressed circRNA only, while  repressed ones were discarded because of their low-expression and unreliability. They identified one differentially-expressed circRNA (circPHC3) through the comparison between ACCs and ACAs. The Authors’ small sample sizes might have contributed to this result due to the low statistical power. Using the validation cohort and RT-PCR assays, the authors found that this circRNA was over-expressed in ACCs and ACAs compared to the normal cortex, while no difference was observed between ACCs and ACAs (although no indication of statistical results). In the validation cohort, the authors further examined three miRNAs, which were among those predicted to be sponged by circPHC3 and are known to be differentially expressed in ACCs. They found some expression differences of these miRNAs in a comparison between ACCs and ACAs, or between ACCs and NACs. However, they did not show the physical evidence that these miRNAs have direct association with circPHC3 and are sponged by it.

Specific comments:

(1)   It is not clear how raw circRNA data were processed by the authors to obtain expression data of the three sample groups and how such data were used to create the heatmap shown in 2. I expect that the authors first calculated fold expression changes by comparing tumor data (ACCs and ACAs) against NACs to obtain their fold changes. Then, these data were compared to create the heatmap between ACCs and ACAs. Alternatively, the authors might have employed several housekeeping genes to obtain the fold expression datasets for ACCs, ACAs and NACs individually. If so, the authors can evaluate differential expression heatmaps not only for ACCs and ACAs, but also for ACCs and NACs, and for ACAs and NACs. Since the authors identified five circRNAs overexpressed in ACCs compared to ACAs in the heatmap analysis, all of them should be included in Figure 2, to show full results obtained with this analysis.

(2)   In RT-PCRs for circRNAs: The authors used external control to calculate dCT values of circPHC3, but this is not an appropriate correction method. For correction of sample difference including samples volumes and variation in sample processing, such as for RNA purification procedures, internal controls should be used, such as several houskeeping circRNAs and average expression levels of all detected circRNAs (in some cases of transcriptome analyses). External controls are employed to monitor and/or adjust the variation occurred during sample processing only. It looks like that the authors did not employ exactly the same controls used in circRNA RT-PCRs as those in the heatmap analysis. This might be one of the reasons why the authors obtained different results in these two analyses.

(3)   Same as in RT-PCRs for miRNAs: Although the authors mentioned their use of both internal and external controls in Text, the Y-axis legend of Figure 4A indicated that only an external control was employed for calculation of dCT values. However, no explanation was provided for “geom.mean of cel-miR-39 and RNU-48 (in Y axis legend)” neither in Text and Figure 4A legend.

(4)   Although several publications indicated differential expression of circRNAs and their associated mRNAs as the authors mentioned, circRNAs and their mRNAs are expressed under the same transcriptional regulation on their sharing coding genes, thus levels of their mRNAs should be demonstrated in tandem with the expression of respective circRNAs

Minor comments:

(1)   The authors should mention in Abstract that they only evaluated “over-expressed” circRNAs.

(2)   Although the authors described in TEXT that ACCs were determined by histological evaluation by two expert pathologists, other standards, such as huge tumor sizes and the presence of distant metastasis and/or local invasion (and associated tumor staging, clear evidence of malignancy), should also be mentioned in Text, not just listing them in Table 1. This is because histological findings do not always predict ACCs as the authors mentioned.

(3)   The list of circRNAs (such as top 10-20 circRNAs) found in comparisons used for creation of the heatmap (Figure 2) should be provided as a supplemental table.

(4)   Appropriate references should be sited for the description in page 9, paragraph 5, lines 312-315.

(5)   One cannot understand the phrase “adj.P values->P values->…” (in page 7, last paragraph, line 265).

Author Response

Please see the attachment. We thank Reviewer 2 for these constructive comments that have clearly improved our manuscript.
